

**Using Eddy Covariance Observations to Determine the Carbon**
**Sequestration Characteristics of Subalpine Forests in the Qinghai-**
**Tibet Plateau**
Niu Zhu[1,2,4], Jinniu Wang[1,2,], Dongliang Luo[3], Xufeng Wang[3], Cheng Shen[1,2], Ning Wu[1]
1   Chengdu Institute of Biology, Chinese Academy of Science, Chengdu 610041,
China
2   Mangkang Ecological Monitoring Station, Tibet Ecological Security Barrier
Ecological Monitoring Network, Qamdo 854500, China
3   Northwest Institute of Eco-environmental Resources, Chinese Academy of
Sciences, Lanzhou 730000, China
4   College of Resources and Environmental Sciences, Gansu Agricultural University,
Lanzhou 730070, China
**Correspondence:** Jinniu Wang (wangjn@cib.ac.cn)





**Abstract:** The subalpine forests in the Qinghai-Tibet Plateau (QTP) act as carbon sinks in the context of climate change and ecosystem dynamics. In this study, we investigated the carbon sequestration function using the *in-situ* observations from an eddy covariance system for the subalpine forests. With two-year contiguous observations, the factors driving the seasonal variations in carbon sequestration potential were quantified. We first revealed the seasonal characteristics of carbon dynamics in the subalpine forests during the growing and dormant seasons, respectively. The diurnal carbon exchange exhibited significant fluctuations, as high as 10.78 μmol $CO_2$ s$^{-1}$ m$^{-2}$ (12:30, autumn). The period from summer to autumn was identified as the peak in carbon sequestration rate in the subalpine forests. Subsequently, we explored the climatic factors influencing the carbon sequestration function. Photosynthetically active radiation (PAR) was found to be a major climatic factor driving the net ecosystem exchange (NEE) within the same season, significantly influencing forest growth and carbon absorption. Increasing altitude negatively impacts carbon absorption at the regional scale and the rising annual temperature significantly enhances carbon uptake, while the average annual precipitation shows a minor effect on NEE. At the annual scale, the observations at the subalpine forests demonstrated a strong carbon sequestration capability, with an average NEE of 389.03 g C m$^{-2}$. Furthermore, we roughly assessed the carbon sequestration status of subalpine forests in the QTP. Despite challenges caused by climate change, these forests possess enormous carbon sequestration potential. Currently, they represent the most robust carbon sequestration ecosystem in the QTP. We conclude that enhancing the protection and management of subalpine forests under future climate change scenarios will positively impact global carbon cycling and contribute to climate change mitigation. Moreover, this study provides essential insights for understanding the carbon cycling mechanism in plateau ecosystems and global carbon balance.

**Keywords:** Subalpine forest; Qinghai-Tibet Plateau; The eddy covariance system; Three Parallel Rivers Region; Carbon sinks

**1 Introduction**

Carbon dioxide ($CO_2$) is a prominent greenhouse gas, and its atmospheric concentration has reached an unprecedented level in recent years, with a recorded peak of 419 parts per million (ppm). Extensive research conducted by numerous scholars has consistently demonstrated that human activities have been the primary catalyst behind the significant surge in atmospheric $CO_2$





concentrations since the 18th century (Stein, 2021). $CO_2$ and $CH_4$ collectively contribute
approximately 70% to the global warming potential among the six greenhouse gases specified in
the Kyoto Protocol (Zhang et al., 2022). As atmospheric $CO_2$ concentrations continue to rise, global
climate warming is gradually intensifying. Therefore, The Paris Agreement urges national
governments to restrict the increase in global average temperature to well below 2.0 ℃ above pre-
industrial levels and to strive to limit it to 1.5 ℃. The increasing atmospheric $CO_2$ levels will lead
to irreversible ecological disasters. For instance, if global consumption of fossil fuels continues to
rise at the current rate, the concentration of $CO_2$ in the atmosphere is projected to double within
approximately 50 years. The rise in temperatures at 80 ℃ latitude could result in the melting of
glaciers, leading to a sea-level rise of 5 m (Mercer, 1978). By the year 2040, most countries are
projected to experience at least one annual disaster with a 50% or higher probability (Fortunato et
al., 2022). Addressing the greenhouse effect caused by carbon dioxide and reducing its impact is a
crucial challenge facing human society today. Reducing regional carbon emissions or per capita
carbon emissions is widely regarded as an effective approach to carbon reduction (Wang et al.,
2023a). Nevertheless, countries around the world have already begun to commit to carbon reduction
and carbon neutrality efforts. On September 22, 2020, during the 75th session of the United Nations
General Assembly, the Chinese government announced "double carbon" goals, which aims to
achieve carbon emission peaking by 2030 and carbon neutrality by 2060, in alignment with
ecological conservation and sustainable development objectives (Yu, 2022). It is predicted that
China's average forest carbon sequestration rate would reach to 0.358 Pg C year$^{-1}$ (petagrams of
carbon per year) by 2060 (Cai et al., 2022). This significant rate of carbon sequestration is expected
to have a substantial impact on the environment and economy, providing negative feedback to global
warming (Pan et al., 2011).

Forests cover approximately 30% of the earth's land surface and store around 90% of the

terrestrial vegetation carbon (Le Quéré et al., 2018). However, currently, there is no method
available to accurately quantify the carbon sequestration potential of forests. Quantitative estimation
of carbon sequestration potential still requires scientists to establish more *in-situ* sites and generate
comprehensive datasets to assess a wide range of area. Initially, individuals' biomass measurements
were used to estimate forest carbon sequestration capacity (Ebermayer, 1876). However, this



method was time-consuming, labor-intensive, and prone to inaccuracies due to the omission of various variables during the calculation process. The development of modeling techniques allowed for the use of simulation methods - forest management models and land ecosystem-climate interaction models, such as the Ecological Assimilation of Land and Climate Observation (EALCO), have been widely applied in this regard (Landsberg and Waring, 1997; Wang et al., 2001). Currently, remote sensing monitoring and the eddy covariance method are widely used. Remote sensing techniques can be used to extract vegetation parameters (such as NDVI) from multispectral bands and estimate the carbon sequestration of entire forests through regression analysis (Laurin et al., 2014). The theoretical foundation of the eddy covariance method was initially proposed by Swibank et al., (Swinbank, 1951). It started to be applied in carbon flux studies of forest ecosystems in the 1980s (Anderson et al., 1984). Nowadays, this method not only accurately measures the carbon exchange between forests and the atmosphere but also integrates other instruments to measure meteorological variables such as light intensity and temperature. It allows for long-term and continuous calculation of carbon flux between forests and the atmosphere. Additionally, it provides fundamental data for establishing and calibrating other models. The eddy covariance method has been widely applied in various ecosystems, including urban areas (Konopka et al., 2021), farmlands (Vote et al., 2015), grasslands (Du et al., 2022a), forests (Kondo et al., 2017), and water bodies (Li et al., 2022).

Net Ecosystem Exchange (NEE) of carbon dioxide is a fundamental parameter in the biogeochemical feedback of the climate system (Graf et al., 2013). The carbon flux in forest ecosystems is influenced by multiple environmental factors. Previous studies have shown that NEE is significantly influenced by photosynthetically active radiation (PAR), air temperature (AT), vapor pressure deficit (VPD), relative humidity (RH), and soil temperature (ST) (Liu et al., 2022). Given the projected future global warming trends, the role of forests as a vast carbon reservoir becomes highly significant and worthy of attention. The Qinghai-Tibet Plateau (QTP) is the highest and largest plateau in the world, with an extensive area of alpine forests covering approximately $2.3 \times 10^5 \, \text{km}^2$. These forests hold tremendous economic and ecological benefits. Since the 1960s, the QTP has experienced a faster warming rate compared to lowland areas. It is projected that this phenomenon will be intensified by the end of the 21[st] century (Li et al., 2019). Currently, the QTP



is considered a weak carbon sink at the overall level, but the carbon source-sink dynamics vary
among different ecosystems (Chen et al., 2022). For instance, most lakes in the QTP are currently
characterized by supersaturated $CO_2$ levels (Cole et al., 1994). Mu et al.(2023), found that the
thermokarst lakes serve as significant carbon sources through carbon flux measurements in 163
thermokarst lakes during the summer and autumn seasons. Wang et al.( 2021), discovered that these
ecosystems act as sinks for carbon dioxide in their study comparing carbon fluxes in ten high-
mountain ecosystems with different grassland types. The alpine meadows in the eastern QTP were
identified as strong carbon sinks, with the highest annual average NEE recorded at -284 g C m$^{-2}$.
Forest ecosystems play a crucial role in the south-eastern edge of the QTP, providing important
support for climate regulation and forestry-based economic activities. However, the QTP is a vast
region, with a widespread distribution of high-altitude and subalpine forests. It is essential for
researchers to conduct long-term monitoring to understand how these forests will respond to climate
change. Furthermore, there is a significant data gap concerning the monitoring of carbon exchange
capacity in the forests of the QTP, indicating the need for further data collection efforts. Based on
this, we have established a carbon flux monitoring site in the subalpine ecosystem of the Three
Parallel Rivers Region, located on the south-eastern edge of the QTP, it lies in the transitional zone
between the Qinghai-Tibet Plateau and the Yunnan-Guizhou Plateau, and is renowned as a global
hotspot for biodiversity (Wang et al., 2022). Our research objectives are as follows:
1)  Determine whether the subalpine forests in the Three Parallel Rivers Region act as a carbon
sink or source, and quantify the annual uptake or release of carbon dioxide.
2)  Investigate the main environmental factors influencing the carbon exchange process in the
subalpine forests and identify the factors with the greatest impact.
3)  Assess the carbon exchange capacity of the subalpine forests in comparison to other
ecosystems of the QTP.
This study will provide a data foundation and background support for accurately estimating
the carbon balance of forests in high-altitude areas and for model simulations in the future.
**2 Materials and Methods**
2.1 Overview of the study site



The study site is located in the Hongla Mountain Yunnan Snub-nosed Monkey National Nature
Reserve in Mangkang County, Tibet, China (29.28633 °N, 98.69096 °E). The elevation of the study
site is 3755 m. The observation period was from November 2020 to October 2022. The study area
experiences large diurnal temperature variations and dry conditions in winter, while the summers
are warm and humid. The climate of the region is characterized as a typical mountainous climate.
The average daily sunshine duration exceeds 10 h, with an annual average temperature of 5℃ and
an average annual precipitation of around 600 mm. The main tree species in the area include *Picea*
*likiangensis var. rubescens, Abies squamata, Sabina tibetica Kom,* and *Abies ernestii.* They are
accompanied by the growth of some *Quercus aquifolioides, Rhododendron lapponicum,* and
*Potentilla fruticosa* shrubs. The vegetation coverage ranges from 70% to 80%, indicating rich
vegetation resources. The dominant soil type is yellow brown soil. The study site is located in the
core area of the Three Parallel Rivers (Nujiang River, Lancang River, and Jinsha River) Region.
The area exhibits a complex and diverse climatic environment influenced by the southwest and
southeast monsoon. The mountainous terrain contributes to distinct vertical climate characteristics
and significant variations in water and heat conditions. The region is characterized by numerous dry
and hot river valleys and widespread distribution of canyons.

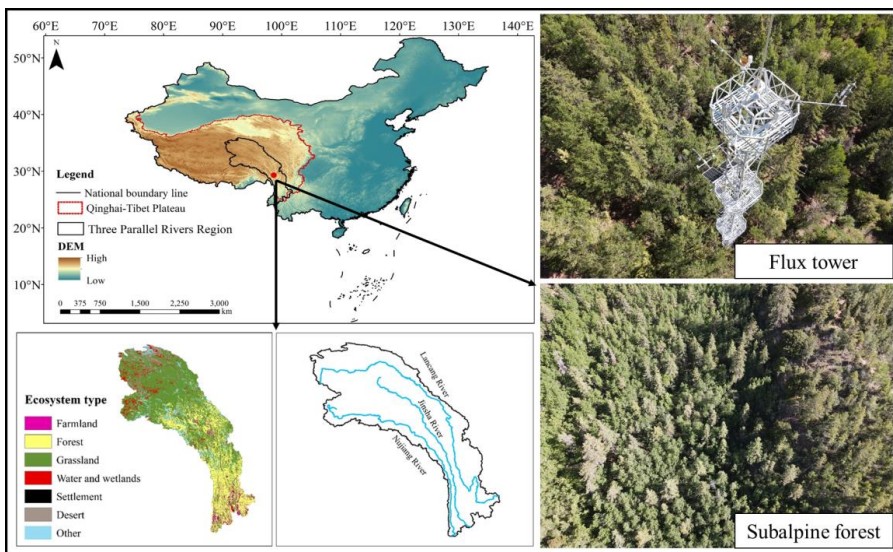

Figure 1 Overview of the study area (The national boundary range in the figure comes from the
http://bzdt.ch.mnr.gov.cn, elevation data from www.gscloud.cn.)



2.2 Eddy covariance system
The flux data in this study were collected from a 35 m-high tower located at the study site. At
the top of the tower, a 3-D wind velocity (Wind Master, Gill, UK) and an open-path infrared
$CO_2/H_2O$ analyzer (LI-7500DS, Li-Cor, USA) were installed to measure $CO_2$ flux. The instruments
had a response frequency of 10 Hz. Additionally, micro-meteorological sensors were placed at
different heights on the tower, including sensors for observing air temperature and humidity
(HMP155A, Vaisala, Finland), soil temperature (TEROS11, LI-Cor, USA), and photosynthetically
active radiation (LI-190R, LI-Cor, USA), among other environmental variables. All data were
recorded at 30-m intervals and stored in a SmartFlux 3 data logger (Li-Cor, USA) for future
download.
2.3 Data processing and quality control
Turbulent transport is the primary form of gas exchange between the near-surface and the
atmosphere. In the case of a homogeneous and flat underlying surface, considering only the
turbulent transport of substances in the vertical direction, the $CO_2$ flux $Fc$ ($\mu$mol m$^{-2}$ s$^{-1}$ or mg m$^{-2}$
s$^{-1}$) within the region can be calculated using the following.
$$F_C = \overline{W'\,CO_2'} \quad (1)$$

Where $W'$ represents the vertical component of 3-D wind speed fluctuations (m/s), $CO_2'$ represents
the fluctuations in measured $CO_2$ mole concentration ($\mu$mol m$^{-3}$), and the overline denotes the
average value over a half-hour time period. A positive value of Fc indicates carbon emissions from
the underlying surface during the given time interval, while a negative value represents carbon
uptake.
The acquired 10 Hz raw data was processed and corrected using the EddyPro software
(EddyPro 7.06, Li-Cor, USA). The correction process involved outlier detection for flux data, lag
elimination, second-order coordinate rotation (Jia et al., 2020), ultrasonic temperature correction
(Schotanus et al., 1983), frequency correction (Moncrieff et al., 1997), and Webb-Pearman-Leuning
(WPL) correction (Leuning and King, 1992). We removed outliers caused by environmental
disturbances such as power outages, rain, snow, and dust particles that interfered with the instrument.
We also corrected errors resulting from non-uniform and non-flat underlying surfaces (Cao et al.,
2019). As a result, we obtained half-hourly flux data with associated data quality indicators. To



evaluate the turbulence steadiness, we employed the "0-1-2" quality assessment method, which
classified flux results into three quality levels: 0 for excellent data quality, 1 for moderate data
quality, and 2 for low data quality (Mauder and Foken, 2011). We removed data points labeled with
a quality level of "2". We further eliminated flux data with negative values during nighttime since
plants do not perform photosynthesis at night. Additionally, we conducted spectral analysis to
identify and remove data points with values significantly deviating from normal. Finally, we utilized
the friction velocity (U*) as a criterion and deleted data recorded during nighttime when U* was
less than 0.28 and 0.39 m s$^{-1}$ (Papale et al., 2006).
NEE of $CO_2$ can be represented by the following:
$$NEE = F_C + F_S \quad (2)$$
Where NEE represents the net ecosystem exchange of $CO_2$, $F_C$ stands for the observed flux during
a specific time period, $F_S$ represents the $CO_2$ storage in the forest canopy, which is assumed to be
zero in this case.
We used the Michaelis-Menten model to fit the daytime NEE ($NEE_{day}$) with respect to PAR to
fill in the missing values during the daytime (Falge et al., 2001):
$$NEE_{day} = \frac{a \cdot PAR \cdot P_{max}}{a \cdot PAR + P_{max}} - R_{day} \quad (3)$$
where: a (μmol $CO_2$/μmol PAR) represents the apparent photosynthetic quantum efficiency, which
characterizes the maximum efficiency of converting light energy during photosynthesis. PAR (μmol
m$^{-2}$ s$^{-1}$) is the photosynthetically active radiation, a measure of the amount of light energy available
for photosynthesis. $P_{max}$ (μmol $CO_2$ m$^{-2}$ s$^{-1}$) is the apparent maximum photosynthetic rate,
representing the maximum $CO_2$ uptake rate under optimal conditions. $R_{day}$ (μmol $CO_2$ m$^{-2}$ s$^{-1}$) is the
daytime dark respiration rate, which denotes the rate of $CO_2$ release during daylight hours. The
parameters a, $P_{max}$, and $R_{day}$ are obtained through non-linear fitting of the Michaelis-Menten model
to the observed data.
During the nighttime, the NEE is modeled using an exponential function of respiration and soil
temperature to fill in the missing values of NEE during the night ($NEE_{night}$) (Lloyd and Taylor, 1994;
Kato et al., 2006):
$$NEE_{night} = a \cdot exp^{(bt)} \quad (4)$$



The parameters $a$ and $b$ are estimated values for the exponential function used in modeling $NEE_{night}$.
The variable $t$ represents the soil temperature measured at the depth of 5 cm. The data processing
software used for this analysis is Origin 2023 (Originlab Corporation, USA). For the missing data,
interpolation was performed using Tovi software (Tovi, Li-Cor, USA) that allows for data
interpolation to fill in the gaps and ensure a continuous dataset for further analysis (Reichstein et
al., 2005).
2.4 Flux splitting

Ecosystem respiration (RE) is the sum of plant and heterotrophic respiration in an ecosystem

and is obtained by adding the measured nighttime data to the extrapolated daytime data. Gross
primary productivity (GPP) is the total amount of organic carbon fixed by green plants through
photosynthesis per unit of time and per unit of area:
$$RE=R_{day}+R_{night} \quad (5)$$
$$GPP=-NEE+RE \quad (6)$$

Carbon use efficiency (CUE) is a crucial parameter that reflects the ability of an ecosystem to

sequester carbon. It is defined as the ratio of net primary productivity to gross primary productivity.
CUE can be expressed using the following equation:
$$CUE=\frac{NEP}{GPP}=\frac{-NEE}{GPP} \quad (7)$$

To study the variation of ecosystem respiration rates with environmental conditions, we

considered the dependence of nocturnal ecosystem respiration on soil temperature (Pavelka et al.,
2007; Mamkin et al., 2023):
$$Q_{10}=\exp(10 \cdot \alpha) \quad (8)$$
$$\ln(NEE_{night})=\alpha \cdot T+\gamma \quad (9)$$
Where T is the soil temperature (°C) and $\gamma$ is an empirical parameter of the equation.
**3 Results**
3.1 Daily average changes in main environmental factors

During the observational period, the environmental conditions exhibited significant

fluctuations. The winter and spring seasons were characterized by cold and dry conditions, while
the summer and autumn seasons were warm and humid. The daily maximum air temperature (AT)
recorded was 15.87 °C (on June 15, 2021), and the minimum temperature was -9.88 °C (on January



17, 2022), with an average of 5.5 °C over the two-year period. The relative humidity (RH) ranged
from a maximum of 93.98% (on August 26, 2021) to a minimum of 6.74% (on April 29, 2021), with
an annual average of 55.89%. The vapor pressure deficit (VPD), which represents the difference
between the saturated vapor pressure and the actual vapor pressure in the air, influences plant
stomatal closure and regulates physiological processes such as transpiration and photosynthesis.
The highest recorded VPD was 1169.8 hPa (on July 5, 2022), and the lowest one was 60.8 hPa (on
August 26, 2021), with an annual average of 446.4 hPa. Soil temperature (ST) exhibited a similar
trend to air temperature and remained relatively stable over short periods. The highest observed soil
temperature was 13.53 °C (on June 27, 2021), while the minimum was -3.78 °C (on January 18,
2022), with an annual average of 6.11 °C. Photosynthetically active radiation (PAR) reached a
maximum value of 779.06 mol m$^{-2}$ s$^{-1}$ (on June 2, 2021), with an annual average of 447.24 mol m$^{-2}$
s$^{-1}$. From March to October, the radiation conditions were favorable for photosynthesis, but
reduction in radiation intensity was observed during rainy, snowy, and cloudy weather conditions.

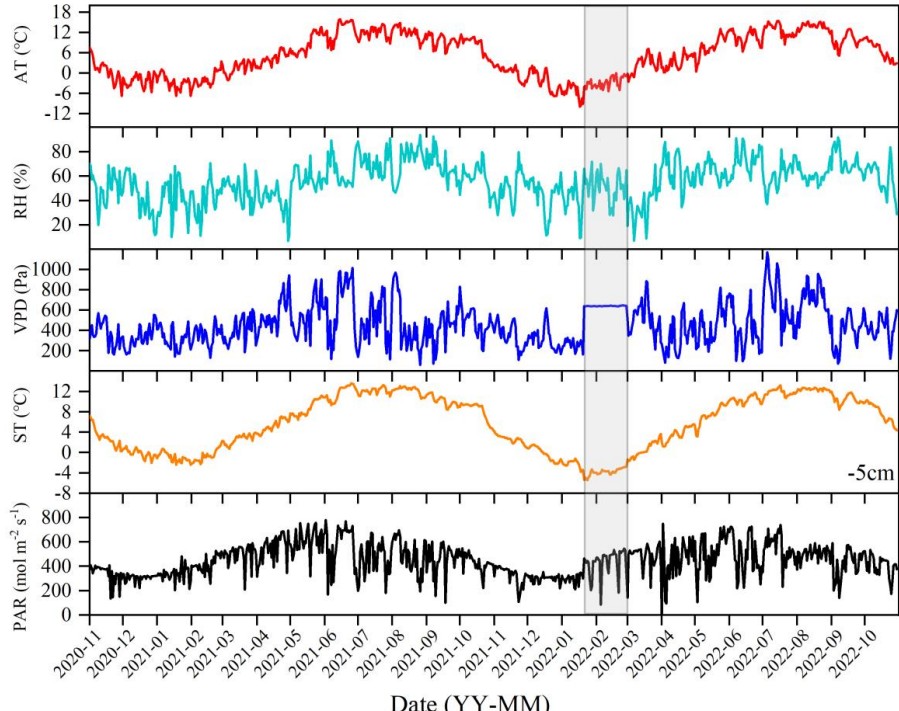




Figure 2 Characteristics of main environmental factors, air temperature (AT), relative humidity
(RH), vapor pressure deficit (VPD), soil temperature (ST), Photosynthetically active radiation
(PAR). (The shaded part of the figure represents the data interpolated by the nearby station)
3.2 The seasonal variations in NEE, RE, and GPP
The observations from the forest ecosystem indicates distinct diurnal and seasonal variations
in NEE and GPP. The NEE and GPP exhibit a pronounced U-shaped curve, with significant seasonal
differences. The summer and autumn are characterized by peak carbon uptake, with the maximum
NEE reaching 10.78 umol $CO_2$ m$^{-2}$ s$^{-1}$ (12:30, autumn). During the nighttime, the ecosystem
generally releases carbon, while during favorable daytime meteorological conditions, it
demonstrates carbon uptake. The peak carbon absorption of the forest ecosystem occurs from 12:00
to 15:00 (Beijing time, UTC+8:00). The carbon sequestration period in summer and autumn is 1.5-
3 hrs longer than in winter. The timing of maximum carbon sequestration capacity changes with
each season. In winter, the transition from nighttime carbon release to daytime carbon uptake occurs
around 08:30, while in summer, it shifts to around 07:30 (Beijing time, UTC+8:00). GPP reflects
the carbon sequestration capacity of the forest, with the recorded daily total productivity highest at
14.76 umol $CO_2$ m$^{-2}$ s$^{-1}$ during summer season of second year, RE exhibits minor diurnal variations
but shows significant seasonal differences, with maximum and minimum diurnal RE values of 0.73
umol $CO_2$ m$^{-2}$ s$^{-1}$ and 0.17 umol $CO_2$ m$^{-2}$ s$^{-1}$, respectively. The respiration rate of the coniferous
forest during the summer and autumn is 5-8 times higher than that in the winter and spring.

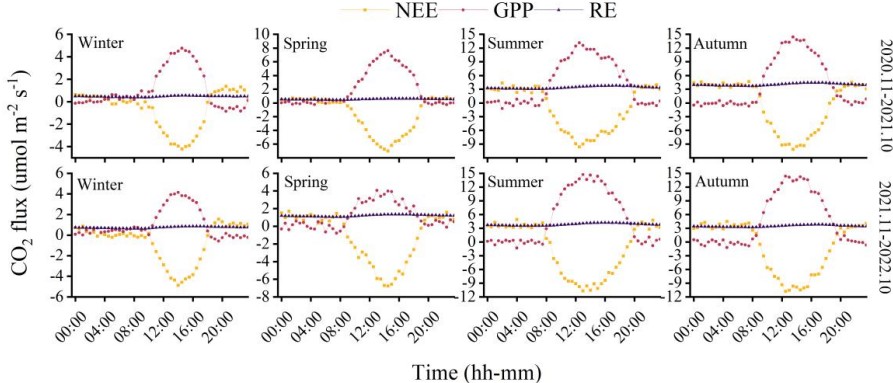


Figure 3 The monthly variations in carbon fluxes
3.3 Relationship between NEE and main environmental factors



The fitting results between NEE and environmental factors indicate that the selected
environmental factors have a significant impact on NEE (*P*<0.001) (Figure 4). However, the
influence of individual environmental factors on NEE varies across different seasons. RH has the
smallest impact on NEE during the summer, while AT, VPD, and PAR exhibit the strongest
influence on NEE during the autumn. These factors consistently have the least impact on NEE
during autumn. In the same season, PAR primarily controls NEE, with an $R^2$ value reaching up to
0.957. Positive values of NEE indicate carbon emissions, while negative values indicate carbon
uptake. Therefore, air temperature, vapor pressure deficit, and PAR all have a significant positive
effect on carbon uptake, while an increase in humidity leads to a noticeable reduction in carbon
uptake.

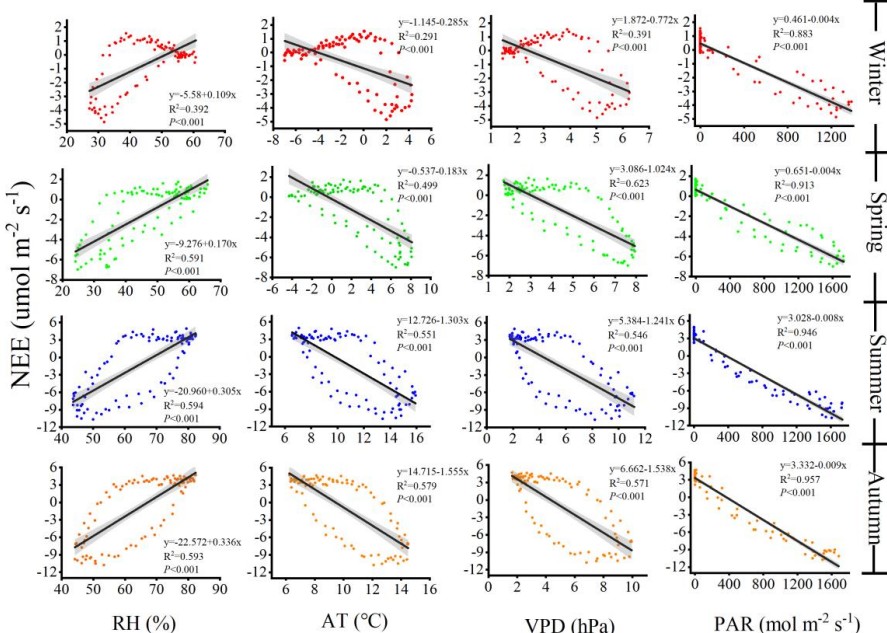


Figure 4 Relationship between NEE and main environmental factors
3.4 Seasonal variation characteristics of NEE, GPP, and RE
The NEE rate did not show significant inter-seasonal differences (Figure 5). However, data
distribution indicates that the variability in NEE rate differs across different seasons, particularly
between the growing seasons (summer, autumn) and the non-growing seasons (winter, spring). The
changes in GPP over the two years were similar, with significant differences observed between the





growing seasons and the non-growing seasons (*P*<0.05). The RE was higher during the growing
seasons compared to the non-growing seasons. The forest ecosystem respiration rate was lowest in
winter and slightly higher in spring. The highest ecosystem respiration occurred in the first year
during autumn, while in the second year, it was highest during summer. This pattern is also reflected
in GPP and NEE.

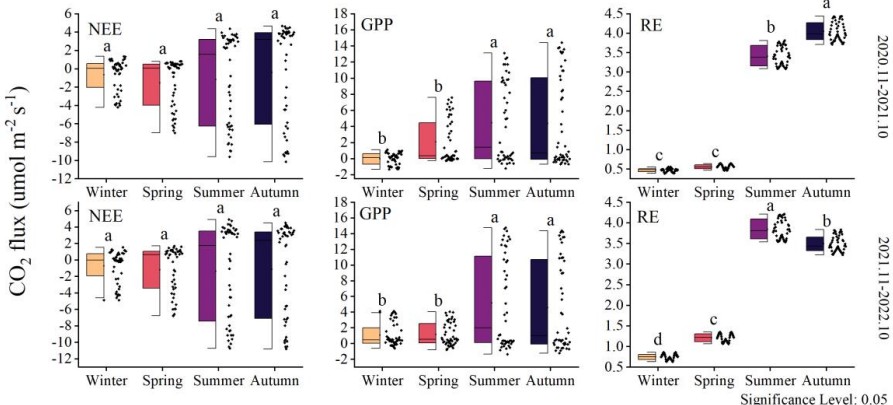


Figure 5 Seasonal variation of carbon fluxes

3.5 Changes in total NEE, GPP, RE, and CUE

The cumulative fluxes over the two years for the forest ecosystem are shown in Figure 6. NEE

indicates the net carbon sequestration in each month. The cumulative respiration reached its highest
value of 363.23 g C m$^{-2}$ in the summer of 2022. The total NEE, GPP, and RE for the first year were
-358.65, 1159.60, and 802.67 g C m$^{-2}$, respectively, and -419.41, 1265.96, and 846.55 g C m$^{-2}$ for
the second year, respectively. The CUE was higher during the cold non-growing seasons and lower
during the growing seasons, with a maximum value of 0.73 and a minimum value of 0.08. The
average CUE over the two years was 0.43 and 0.41, respectively.

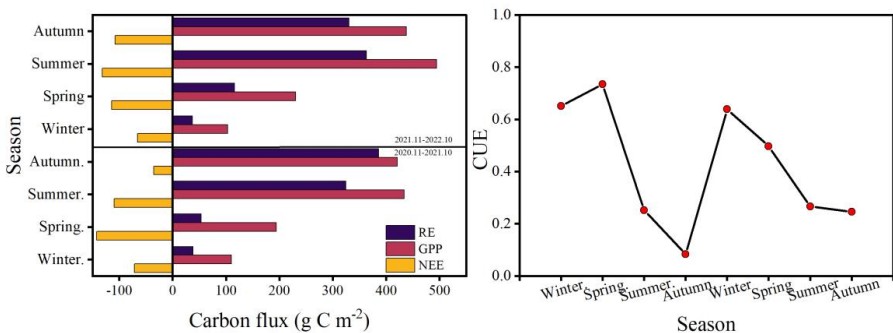






303     Figure 6 Change in total carbon flux and carbon use efficiency

304 3.6 The carbon sequestration potential of subalpine forests of QTP

305 To clarify the carbon sequestration contribution of the subalpine forests found on the QTP, we

306 collected and compared 83 research results from 49 studies that utilized EC systems (Figure 7).

307 Ecosystems with high vegetation cover exhibited higher annual cumulative carbon sequestration.

308 Among these ecosystems, the subalpine forests on the QTP showed the highest carbon sequestration

309 potential, reaching an average of 391.48 g C m$^{-2}$ per year. The carbon sequestration potential of

310 different ecosystems ranked as follows: forest > meadow > steppe > shrub. The average value for

311 wetlands indicated that they are a significant source of $CO_2$, releasing 56.93 g C m$^{-2}$ into the

312 atmosphere annually.

313 We also analyzed the influence of elevation, mean annual temperature, and precipitation on

314 NEE at these sites in the QTP. It was found that increasing elevation had a negative impact on

315 carbon uptake, while higher mean annual temperatures significantly increased carbon uptake. Mean

316 annual precipitation had a weak influence on NEE. These findings highlight the important role of

317 subalpine forests in carbon sequestration in the QTP and provide insights into the factors that affect

318 carbon exchange on the QTP, such as altitude, temperature, and precipitation.

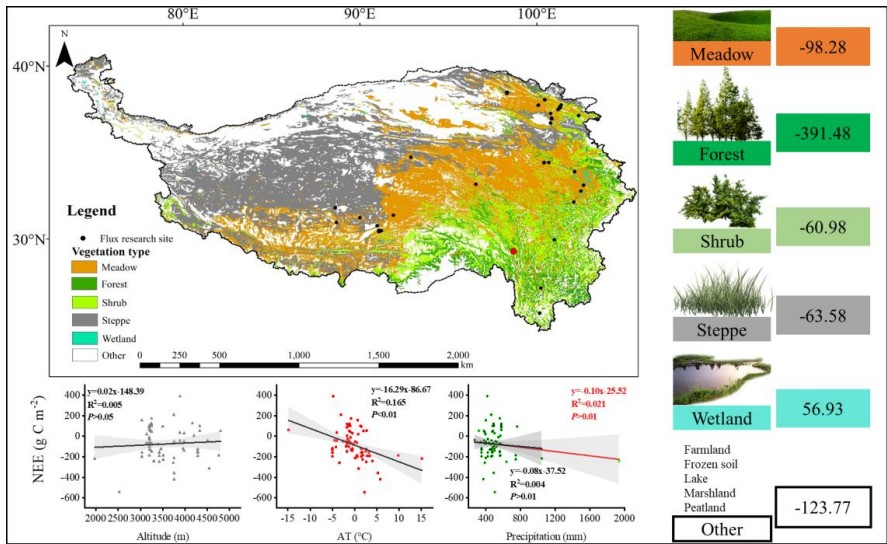

320 Figure 7 Carbon exchange potential of different ecosystems in the Qinghai-Tibet Plateau

321 **4 Discussion**



4.1 Main factors affecting the carbon sequestration function of subalpine forests

Climate change is the significant factor affecting the vegetation's carbon sequestration capacity, particularly at the seasonal scale due to phenological changes (Acosta-Hernández et al., 2020). Our study has demonstrated that, in the short term, NEE is primarily influenced by factors such as PAR, AT, RH, and VPD. These factors play a role in regulating vegetation photosynthesis and, consequently, carbon uptake. For instance, PAR represents the portion of solar energy that can be utilized by plants and is an essential component in chloroplast reactions. AT regulates the activity of enzymes involved in light and dark reactions, which may contribute to seasonal variations in NEE. RH and VPD impact the entire process of photosynthesis by influencing the concentration of $CO_2$ in the air and the stomatal conductance (the pathway for $CO_2$ exchange). In different seasons, the same influencing factors exhibit varying degrees of contribution to NEE. For example, during winter, when the climatic conditions are relatively harsh with low air temperature and humidity, the forest maintains a low level of carbon uptake. While the forest continues to absorb carbon dioxide, the uptake remains limited at a low level under such unfavorable conditions. On longer time scales, such as annual and decadal variations, the inherent changes in forest NEE may be attributed to disturbances and recovery (Hayek et al., 2018). Research by Amiro (2001) has demonstrated that disturbances caused by fire and logging have been found to regulate the carbon balance of northern forests in Canada over several decades. Additionally, there are close relationships between subtle climate changes, stand dynamics, tree age, post-disturbance time, and forest carbon storage and cycling (Bradford et al., 2008). Compared to naturally regenerating forests, actively restored forests exhibit higher rates of carbon accumulation. Restoration efforts have been shown to increase aboveground carbon density recovery rates by more than 50% over a decade, from 2.9 to 4.4 megagrams per hectare per year (Philipson et al., 2020). The carbon dioxide generated by soil microbial activity is an essential component of forest ecosystem respiration. Soils contain the largest organic carbon reservoir on Earth, three times more than the carbon content in the atmosphere (Tifafi et al., 2018). With climate warming, soil microorganisms, and root systems will decompose soil organic carbon at a faster rate, releasing carbon dioxide into the atmosphere more rapidly. Temperature plays a more sensitive role in soil carbon turnover in cold climate regions compared to warmer conditions (Koven et al., 2017). Ecological respiration sensitivity to temperature is

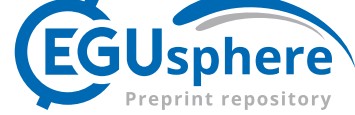

represented by the $Q_{10}$ coefficient. In this study, seasonal variations influenced the magnitude of
$Q_{10}$ (as shown in Figure 8). The calculated $Q_{10}$ values for each season are as follows: 9.025, 2.22,
2.71, and 4.48. The winter season exhibited the highest sensitivity of forest ecosystem respiration
to temperature, indicating that respiration rates in the winter are more responsive to changes in
temperature compared to other seasons.

Our integrated analysis (as shown in Figure 7) reveals that, despite the high elevation of the

"Third Pole", the topographic factor of elevation does not have a significant impact on carbon uptake.
Instead, NEE gradually increases with a steep rise in elevation. Research conducted by WANG et
al.(2023b), indicates that average temperature and precipitation are the main driving factors of
interannual variations in NEE in alpine meadows and alpine grasslands. Decreased precipitation can
cause some regions of high precipitation-dependent alpine grasslands to transition into carbon
sources. It is worth noting that, among all data collection sites, alpine wetlands show an average
carbon source trend. Due to prolonged flooding and low temperatures, microbial activity in alpine
wetlands is hindered, and the accumulation of organic carbon from plant litter decomposition is
substantial. As a result, approximately 56.93 g C m$^{-2}$ is emitted into the atmosphere annually.
Previous studies have indicated that NEE in alpine wetlands is increasing with global warming
(Yasin et al., 2022).

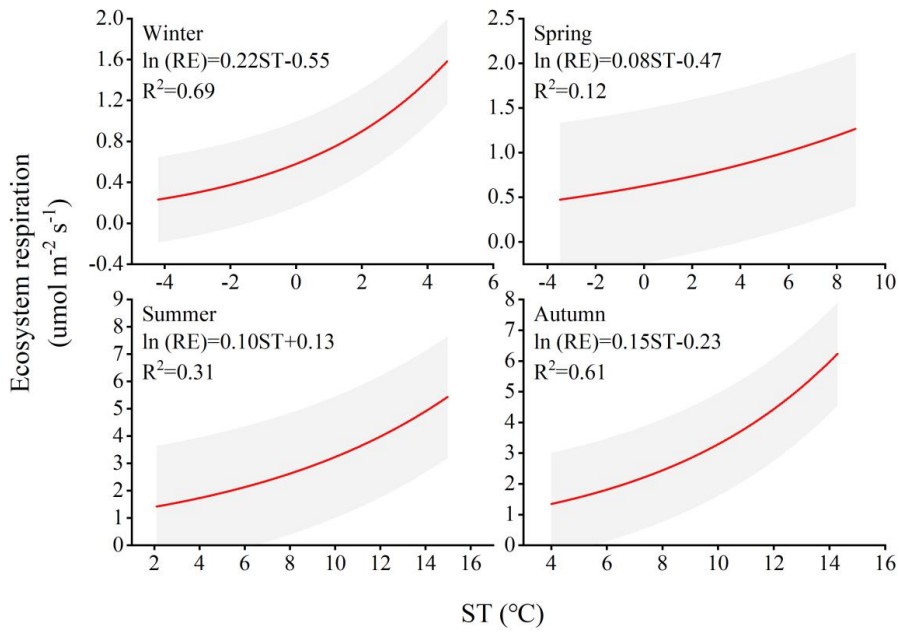




Figure 8 Relationship between NEEnight and 5 cm depth soil temperature in different seasons
4.2 Sustained carbon sequestration of subalpine forests

Subalpine forests are integral components of global alpine ecosystems and play crucial roles

in the global carbon cycle. It is worth noting that atmospheric carbon dioxide levels are steadily
increasing due to human activities. Mitigation of greenhouse gas emissions to achieve carbon
neutrality through natural processes is undoubtedly the most cost-effective and convenient option,
in addition to industrial carbon reduction measures. Therefore, understanding whether natural
ecosystems possess sustained carbon sequestration functionality is of utmost importance. Tian.
(2018), used a structural-dynamic approach to predict that US forests will continue to sequester
carbon for majority of the next century, sequestering 128 Tg C year$^{-1}$. Consistent with our findings,
our study on subalpine forests demonstrates that they continue to absorb carbon dioxide even during
winter, which aligns well with measurements taken in the vicinity of Mount Fuji in Japan
(Mizoguchi et al., 2012). Furthermore, their research further confirms that northern forests exhibit
higher carbon uptake capacity. The age of subalpine forests is a crucial factor influencing sustained
carbon sequestration. Based on NPP simulations of natural subalpine forests in the Northern Rockies,
Carey. (2001), found that aboveground net primary productivity reaches its maximum after
approximately 250 years, followed by a decline. This challenges previous notions regarding the
carbon sequestration potential of forests that are approximately over 100 years of age. In our study,
the subalpine forest exhibited a sparse shrub understory, indicating that it is not a mature forest
ecosystem. This may be a significant factor contributing to its stronger carbon sequestration capacity
compared to the high-mountain forests (mature forests) in Mount Gongga in the QTP (Yuanyuan et
al., 2018). However, its carbon sequestration ability is slightly weaker than that of the Qilian
Mountains high-mountain forests (approximately 60-70 years old) in the QTP (Yuanyuan et al.,
2018; Du et al., 2022b). Although existing flux monitoring results of high-altitude forests in the
QTP indicate that these forest ecosystems act as carbon sinks, it is important to consider that globally
there are still many cold regions where coniferous forests serve as carbon sources. For example,
continuous $CO_2$ flux monitoring from native boreal forests in Sweden for over 10 years indicates
that they are a net carbon source, this is attributed to the contribution of woody debris to RE due to
disturbances such as extreme weather events, fires, insect infestations, and pathogen attacks



(Hadden and Grelle, 2017). In the summer of 2018, Europe experienced a heatwave that affected
the carbon cycling in forests. The southern Estonian mixed coniferous-deciduous forest, under the
influence of the heatwave, transitioned from a net carbon sink to a net carbon source in 2018
(Krasnova et al., 2022). Particular attention should be paid to the long-term monitoring in high-
altitude environments of the impact of disturbances on forest carbon sequestration capacity. Our
study has shown that forests in the QTP have the strongest carbon sink capacity, indicating that
alpine forests will have an important sustained effect on carbon reduction in the QTP in the context
of future climate change, but whether this sustained effect will be longer than other ecosystems is
still unknown. However, a modeling experiment in a large semi-arid area of California predicted
that grasslands are more resilient carbon sinks than forests in responding to climate change in the
21st century (Dass et al., 2018). In terms of carbon sequestration rate, forests on the QTP were
significantly stronger than other ecosystems, followed by grasslands, while alpine deserts and alpine
grasslands in the north-western and southern regions were the main carbon sources (Wu et al., 2022).
Forests are mostly distributed in the south-eastern margin of the QTP and the mid-altitude area near
3000 m in the Sichuan-Tibet alpine gorge area, with an area of $19.3 \times 10^4$ km$^2$ (Y et al., 2022).
Based on the average value of a few current carbon flux monitoring, the forest in the QTP will
absorb about $75.5 \times 10^5$ T C year$^{-1}$.
**5 Conclusion**
This study explores the carbon sequestration function, seasonal variations, and climate drivers
of subalpine forests in the QTP. Over the observational period, environmental factors exhibited
significant fluctuations, with cold and dry conditions prevailing during winter and spring, while
warm and moist conditions characterized the summer and autumn seasons. The research reveals that
the subalpine forest possesses enormous carbon sequestration potential, with a total NEE, GPP, and
RE of -358.65, 1159.60, and 802.67 g C m$^{-2}$, respectively, and -419.41, 1265.96, and 846.55 g C m$^{-2}$
for two years, respectively. Individual environmental factors exhibited varying effects on NEE in
different seasons, with relative humidity having the least impact on NEE during summer, while air
temperature, saturated vapor pressure deficit, and photosynthetically active radiation had the most
significant influence during autumn, with minimal effects from these factors during other seasons.
Moreover, photosynthetically active radiation was identified as the primary control of NEE in the



same season. The NEE rate did not exhibit significant differences across seasons. Combining results
from other eddy covariance sites on the QTP, this study highlights those forests have the highest
carbon sequestration potential, reaching 391.48g C m$^{-2}$ annually, followed by meadows, steppes,
and shrubs. Wetlands, however, were identified as substantial carbon dioxide source. Despite the
challenges posed by climate change, the subalpine forests in the QTP retain substantial carbon
sequestration potential. Strengthening conservation and management efforts for subalpine forests is
crucial to ensure their continued and significant carbon sequestration function in the future. Overall,
this research underscores the vital role of subalpine forests in the QTP as essential carbon sink
regions, playing a critical role in the context of global climate change.
***Data availability.*** The data are available from the authors on request.
***Authorship contributions.*** **Niu Zhu:** Conceptualization, study design, data analyses,
visualization, writing-original draft. **JinNiu Wang:** study design, writing—review & editing,
supervision, project administration, funding acquisition. **Dongliang Luo and Xufeng Wang:**
writing-reviewing & editing. **Cheng Shen and Ning Wu:** resources, data curation, supervision. all
authors approved the final manuscript.
***Declaration of competing interest.*** The authors declare that they have no conflict of interest.
***Acknowledgements.*** We thank Ms. Neha Bisht for her substantial comments and language
revision on improving the manuscript. This study was funded by CAS "Light of West China"
Program (2021XBZG-XBQNXZ-A-007); National Natural Science Foundation of China
(31971436); State Key Laboratory of Cryospheric Science, Northwest Institute of Eco-Environment
and Resources, Chinese Academy Sciences (SKLCS-OP-2021-06).

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
