# Peer review of "Using Eddy Covariance Observations to Determine the Carbon Sequestration Characteristics of Subalpine Forests in the Qinghai- Tibet Plateau"

_EGUsphere, 2023_

## Author Comment (AC2)

Dear Reviewer,

We would like to express our sincere gratitude for dedicating considerable time and effort to providing valuable feedback on our manuscript. Your insightful comments and suggestions have proven instrumental in enhancing the overall quality of our work. We have meticulously considered and incorporated your feedback into our revised manuscript.

1.Some extra data quality control on the eddy covariance (EC) measurements is still needed, and much key information was still missing. Firstly, in lines 180-184, it is not clear for me how the authors discard nighttime NEE data that were observed when friction velocity (u*) is less than 0.28 and 0.39 m s$^{-1}$. More importantly, there is no such u* criteria of 0.28 and 0.39 m s$^{-1}$ in the study of Papale et al., 2006. Besides, based on my understanding in Eddy Covariance data processing, the "0-1-2" labels (which were mistakenly interpreted as a method to evaluate the turbulence steadiness, in line 177) are not enough for the quality control, extra steps such as the median of absolute deviation about the median (MAD) method from Papale et al., 2006 should be applied to detect the outliers as well. In addition, the number of measurements that were discarded in each outlier detection should be revealed in order to evaluate the quality of the EC dataset. Secondly, the footprint analysis results should be stated in the manuscript. At this stage, key information about the underlying terrain, the composition of each tree species in the forest, the age of the forest, and the footprint of the EC tower are still missing. Finally, in Lines 188-189, it could be problematic to set the storage flux as zero. Since continuous concentration profile measurements are lacking in this study, I suggest the authors apply the decoupling filtering method (Thomas, C.K., Martin, J.G., Law, B.E., Davis, K., 2013. Toward biologically meaningful net carbon exchange estimates for tall, dense canopies: multi-level eddy covariance observations and canopy coupling regimes in a mature Douglas-fir forest in Oregon. Agric. For. Meteorol. 173, 14–27.) to account for both the storage and advection effects.

**Response**:Regarding the uncertainty in the removal of nighttime NEE data based on friction velocity (u*), we appreciate your clarification. In our data processing using the TOVI software, we employed the moving point method (MPT) to calculate u* thresholds separately for each year, aiming to mitigate potential errors due to prolonged computations. We acknowledge the error in referencing Papale et al., 2006, and have duly corrected it in the revised manuscript. Additionally, we have provided a more accurate description of our data processing methods and the criteria for discarding weak turbulence periods. Concerning the "0-1-2" labels, we agree with your suggestion. Prior to this step, multiple quality controls, utilizing Eddypro and TOVI, were conducted, resulting in 10Hz data with quality indicators. We have clarified the rationale behind removing data labeled as "2" during the final outlier detection process. Recognizing the importance of footprint analysis, we have supplemented Table 1 in the manuscript with information on the distances of different flux contribution areas from our flux tower during different seasons.

Table 1 Distance of flux contribution area

| Year | 2020.11-2021.10 | | | | | 2021.11-2022.10 | | | | |
| --- | --- | --- | --- | --- | --- | --- | --- | --- | --- | --- |
| Season | Winter | Spring | Sunmmer | Autumn | Yearly | Winter | Spring | Sunmmer | Autumn | Yearly |
| 10% flux contribution distance (m) | 44.9 | 49.4 | 47.9 | 43.2 | 46.35 | 48.4 | 44 | 43.4 | 42 | 44.5 |
| 30% flux contribution distance (m) | 107.8 | 112.3 | 107.5 | 99.3 | 106.7 | 109.5 | 101.9 | 103.8 | 100.9 | 104 |
| 50% flux contribution distance (m) | 162.7 | 163.1 | 160.8 | 151.2 | 159.5 | 168.6 | 148.6 | 160.6 | 153.9 | 157.9 |
| 70% flux contribution distance (m) | 228.9 | 219.5 | 235 | 220.1 | 225.9 | 239 | 217 | 235.3 | 217.6 | 227.2 |
| 90% flux contribution distance (m) | 337.4 | 361.2 | 342.2 | 344 | 346.2 | 370.4 | 355.2 | 357.9 | 344.8 | 357.1 |

We acknowledge the significance of providing key information about the underlying terrain, forest composition, and age of the forest. We have addressed these gaps in the manuscript, explaining the slope of the underlying terrain and providing details about the forest composition and age. The study site is located in the Hongla Mountain Yunnan Snub-nosed Monkey National Nature Reserve, characterized by trees below 30m, minimal human disturbance, and active growth due to ecological restoration efforts by the local government, including the planting of a small number of birch trees (2.1 Overview of the study site).

We appreciate your suggestion regarding the treatment of storage flux. In response, we have modified our description to acknowledge the potential influence of storage flux and incorporated the decoupling filtering method from Thomas et al., 2013, to account for both storage and advection effects.

2.The introduction and discussion part seem to me a bit plain. Both the importance of subalpine forests in the QTP and its correlations with climate change factors in regard to NEE, GPP, and ER are not deeply revealed and discussed. I recommend the authors to elaborate more on the degradation of permafrost in the QTP and the coming upward migration of tree line. In addition, only linear regression was used to explain the links between NEE and environmental factors. It is highly recommended that the authors utilizing other advanced methods such as PCA loading or wavelet analysis to reveal the details in these correlations.

**Response**:Thank you for your valuable suggestions. Your insights have been crucial in improving the introduction and discussion sections. We have emphasized the importance of subalpine forests in the Qinghai-Tibet Plateau and their correlations with climate change factors, addressing the perceived plainness. Furthermore, we have expanded the discussion on permafrost degradation in the Qinghai-Tibet Plateau and the anticipated upward migration of the tree line to enhance the manuscript's depth and richness. Regarding your second point, we agree with the need for advanced methods in correlation analysis. In response, we have incorporated Principal Component Analysis (PCA) into the manuscript to provide a more nuanced perspective on the relationships between NEE and environmental factors.

3.The division of seasons is ambiguous to me. In Lines 18-19, the authors used the term growing and dormant season. However, the length of the growing season and dormant season is missing, making it hard to follow what the authors are trying to describe. As for the normal four seasons, I

assume the authors used the common calendar which delineates them into three months each season. This might be not useful for evaluating the seasonality of NEE since the vegetation status was not illustrated. I would suggest the authors carefully define the length of growing and dormant seasons based on relevant conditions such as air temperature and soil moisture and then analyze the seasonality of NEE and its climatic controls.

**Response:** We appreciate your keen observations regarding the ambiguity in the division of seasons in our manuscript. We acknowledge the inadequacy of using the common calendar to describe the growing and dormant seasons without conducting a detailed analysis in the current study. In response to your suggestion, we have removed the inappropriate descriptions related to growing and dormant seasons from the manuscript. Additionally, we understand the importance of accurately defining the length of seasons based on relevant conditions such as air temperature and soil moisture. However, given the limited duration of our monitoring (two years) and the absence of explicit research results in our study area, using the common calendar was chosen to avoid potential errors in season delineation. We recognize the limitations and acknowledge the need for more precise definitions. Furthermore, we are actively addressing the issue of seasonality by deploying multiple phenocams in the forest. These cameras are continuously monitoring phenological dynamics, and we plan to integrate long-term environmental factor data, such as air temperature and soil moisture, in future research to accurately define seasonal boundaries based on empirical data.

4.The comparison of NEE measurements from ecosystems over the QTP seems redundant to me. Even if the compilation is needed for this study, details such as the number of each ecosystem, the year of the observation, and the general environmental factors (air temperature, precipitation) should be illustrated clearly. It is also important to explain how the average NEE of each ecosystem was calculated and how these sites can represent the same kind.

**Response :** Thank you for your insightful comments regarding the comparison of NEE measurements from ecosystems over the Qinghai-Tibet Plateau in our manuscript. We acknowledge the sudden introduction of this section in the text and recognize the need for more context in the preceding sections, which we will emphasize in the revised manuscript. The comparison of carbon sequestration in alpine forests with other ecosystems on the Qinghai-Tibet Plateau is a crucial objective of our study. Previous research on the carbon sink potential of alpine forests in the Qinghai-Tibet Plateau has been limited, and our study aims to fill this knowledge gap. We appreciate your acknowledgment of the importance of this analysis. Furthermore, we appreciate your suggestion to provide additional details in this section. We have made careful revisions to address this concern. Specifically, we have increased the information on the number of collected ecosystem types and provided a more comprehensive description of the environmental factors, including the temperature and precipitation ranges for each ecosystem. Detailed information such as vegetation types, latitude, and longitude for each site will be compiled in an Excel file, which will be made available as supplementary material to meet the readers' requirements.

5. Line 29, the NEE value should be negative.
**Response:** We have added a negative sign in front of the NEE value.

6. Line 30, rephrase the word "enormous" if you can't support the argument with statistics.
**Response:** Our initial description was inappropriate, and we have replaced the word. The revised

sentence now reads, "Despite challenges caused by climate change, these forests still have the potential to sequester carbon."

7. Lines 39-40, references are needed to show where this number of 419 comes from and which year was the measurement.
**Response:** We have added references to support the mentioned number, including the relevant year of measurement.

8. Lines 67-68, this statement seems too arbitrary and could be controversial.
**Response:** We have revised the statement to convey a more accurate representation. The updated sentence now reads, "However, currently, there are various methods available to accurately quantify the carbon sequestration potential of forests, each with its own advantages and disadvantages."

9. Lines 80-89, the description and review here seem too simple and redundant to me.
**Response:** We have refined the description to make it more concise. The revised content is, "The eddy covariance (EC) method allows continuous, long-term carbon flux calculation, providing fundamental data for model establishment and calibration. It is widely applied across various ecosystems, including urban areas, farmlands, grasslands, forests, and water bodies (Konopka et al., 2021; Vote et al., 2015; Du et al., 2022a; Kondo et al., 2017; Li et al., 2022)."

10. Line 133, delete this sentence.
**Response:** We have removed this sentence.

11. Lines 134-135, add relevant references to show where these measurements come from.
**Response:** We have added references at this point to support the mentioned measurements.

12. Line 139, be careful when using the term "vegetation resources", would be better to be more specific.
**Response:** We have removed the sentence containing "vegetation resources."

13. Lines 141-144, refine these sentences to focus more on the subalpine forest ecosystem.
Response: We have added more specific descriptions focusing on the subalpine forest ecosystem.

14. Line 147, the source of the ecosystem type map needs to be reported as well.
**Response:** We have indicated the source of the ecosystem type map.

15. Line 152, it is the frequency of measurements rather than the response frequency.
**Response:** We have used "frequency of measurements" instead of "response frequency."

16. Line 153, the specific heights need to be revealed here.
**Response:** We have added specific heights in the manuscript.

17. Lines 159-168, these basic eddy covariance descriptions should be more concise.
**Response:** We have condensed these sentences to make them more concise.

18. Line 169, should be calibration rather than correction.
**Response:** We have replaced "correction" with "calibration."

19. Line 170, please confirm whether EddyPro has the function of outlier detection.
**Response:** We have confirmed that the processing software used has the function of outlier detection.

20. Lines 175-176, please elaborate on how this process was applied to the correction.
**Response:** We have provided additional details on how this process was applied for correction.

21.Lines 190-191, this is the gap-filling strategy, not filling the missing value.
**Response:** We have corrected this inaccurate description.

22.Line 199, replace "a" with "α".
**Response:** We have replaced "a" with "α".

23.Line 201, ecosystem respiration.
**Response:** We have used "ecosystem respiration" instead of "respiration".

24.Line 2017-210, are both daytime and nighttime data gaps being filled using the Tovi software? The number of the gaps should be stated.
**Response:** We have added information on data completeness during the data processing.

25.Line 211, we normally use the term "flux partitioning", not "flux splitting".
**Response:** We have used "flux partitioning" instead of "flux splitting".

26.Line 222, what are the environmental conditions?
**Response:** We have used "factors" instead of "conditions".

27.Lines 236-238, delete this sentence.
**Response:** We have removed this sentence.

28.Line 241, what is 'short periods'? Please be precise.
**Response:** We have removed this inaccurate description from the manuscript.

29.In figure 2, the unit of VPD should be hPa rather than Pa.
**Response:** We have corrected the unit of VPD in Figure 2.

30.Line 250, where is this nearby station, and which data was interpolated?
**Response:** We have added the name and coordinates of the nearby station in the manuscript.

31.Line 255 and Line 258, the UTC+8 time needed to be revealed in the first place.
**Response:** We have placed the UTC+8 time at the beginning of the mentioned lines.

32.Line 258, the term "carbon sequestration period" is not defined.
**Response:** We have removed "period".

33.Lines 261-265, rephrase this sentence.
**Response:** We have rephrased this sentence.

34.Line 265, only one unit is required in this sentence.
**Response:** We have modified this section for clarity.

35.Lines 270-271, the P value is not enough to determine the significance.
**Response:** We have revised Figure 3, utilizing PCA for further analysis.

36.Line 276-277, delete this sentence.
**Response:** We have deleted this content.

37.In figure 4, please confirm whether the unit of VPD is correct or not.
**Response:** We have confirmed the unit in Figure 4.

38.Line 281, Figure 4. Relationship…
**Response:** We have modified this expression and added a new figure.

39.Line 283, delete "rate".
**Response:** We have removed "rate".

40.Lines 313-318, more discussion and description are needed to support your claim of the "findings". This writing style fits the conclusion part, not the result part.
**Response:** We have modified this part for clearer result presentation and added more detailed information.

41.Line 350, what is "ecological respiration sensitivity"?
**Response:** We have used "ecosystem" instead of "ecological".

42.Lines 386-388, more evidence is needed to support this conclusion.
**Response:** We have removed this speculative statement as it lacked detailed ecosystem information.

43.Line 412, please confirm whether this reference style here is correct or not.
**Response:** Thank you for pointing out the issue. We have made the necessary corrections.

44.It would be better if the authors could show the standard deviation error bars in figure 3.
**Response:** You have provided an excellent suggestion, and we have added the standard deviation error bars to Figure 3.

Our heartfelt thanks go to the esteemed reviewer for their dedicated efforts and valuable insights that have immensely enriched our manuscript. The meticulous attention to detail and thoughtful

comments have played a pivotal role in elevating the overall quality of our research. We sincerely appreciate the time and expertise invested by the reviewer, which have undoubtedly contributed to the refinement and clarity of our work. Your commitment to excellence in the peer review process has been a guiding light, and we are truly grateful for the collaborative spirit demonstrated throughout this constructive review.

---

## Author Comment (AC3)

Dear Reviewer,

Thank you for your valuable feedback on our manuscript. We appreciate your constructive comments that have been instrumental in refining the manuscript. It has improved the clarity and accuracy of our study on subalpine forests in the Qinghai-Tibet Plateau. Below, we address each of your specific comments and modifications made in response:

Specific comments:

L14-15 The abstract starts with "The subalpine forests in the Qinghai-Tibet Plateau act as carbon sinks in the context of climate change and ecosystem dynamics. " I think the tone should be adjusted because we cannot conclude this point before we conducted this work.

**Response:** Thank you for the reviewers comment. We appreciate the point raised, and we have incorporated the suggested change into the manuscript. The revised sentence now reads: "The subalpine forests of the Qinghai-Tibet Plateau are one of the crucial components in the carbon cycling system in the context of climate change and ecosystem dynamics."

L20-21 The time of NEE reached the peaks, in fact, is not stable. So, I think the time (as high as 10.78 μmol $CO_2$ $s^{-1}$ $m^{-2}$ (12:30, autumn)) here has no meaning.

**Response:** Certainly, your point about the instability of the time when NEE reaches its peaks is valid. We acknowledge that including this information in the abstract is inappropriate. Therefore, we have removed this description from the abstract.

L25-26 As the study is based on the site-level EC observations, we cannot directly conclude "Increasing altitude negatively impacts carbon absorption at the regional scale".

**Response:** We have removed the inappropriate conclusion from the abstract, as it is not directly supported by the site-level EC observations. Additionally, in the analysis of Figure 7, we conservatively stated that the elevation of these sites "may influence NEE" across the entire Qinghai-Tibet Plateau.

L29 Add the time unit to NEE.

**Response:** Thank you for your suggestion. We have added the time unit to NEE in the manuscript.

L92-94 The variability in NEE is not affected by climate factors, but also influenced by the biotic factor such as NDVI or LAI. The work by Tang et al. 2022, Forest Ecology and Management can be supplemented here.

**Response:** Thank you for pointing out the shortcomings in the introduction. We have addressed this by adding the following content: "Furthermore, research suggests that the NEE is influenced by biotic factors such as NDVI (Normalized Difference Vegetation Index) and LAI (Leaf Area Index)," referencing the work by Tang et al. 2022 in Forest Ecology and Management.

L106 Delete "in their study".

**Response:** We have removed the phrase "in their study" from this paragraph.

L219 It is defined as the ratio of net primary productivity to gross primary productivity. Here, NEP

is not net primary productivity (NPP). Need to revise.

**Response:** Thank you for bringing this to our attention. We have corrected the error, and the revised sentence now reads: "It is defined as the ratio of net ecosystem productivity (NEP) to gross primary productivity."

L239-240 The highest recorded VPD was 1169.8 hPa (on July 5, 2022), and the lowest one was 60.8 hPa (on August 26, 2021), with an annual average of 446.4 hPa. I think the unit of VPD is wrong here. Please also see the figure.

**Response:** Thank you for your feedback. We have addressed the unit error in this paragraph and corrected it.

Figure 3 The diurnal dynamics of NEE, GPP and Re seems surprised. For example, Re nearly kept a line across seasons. I also cannot understand the forest remains to be strong carbon sink in winter. I think the GPP should be zero as the temperature is below zero.

**Response:** Thank you for your insightful comments. Regarding your concerns, we would like to provide the following explanation: Our monitoring results indicate that the diurnal dynamics of NEE, GPP, and RE exhibit significant variations, but when observed at a seasonal scale, they also show considerable differences, as illustrated in Figure 5. During winter, carbon flux values are much smaller compared to other seasons. Despite the fact that the daily average temperature in winter is often below zero, there is a wide range of temperature fluctuations within a day (as reflected in Figure 4). Daytime temperatures generally remain above 0°C, indicating that the subalpine forest is still undergoing weak photosynthesis and is not completely dormant. Although NEE, GPP, and RE are much smaller in winter, the results show that their seasonal averages are greater than zero. The predominant tree species in this forest are conifers, which allows the forest to maintain a weak carbon sink even during the winter months.

L327-328 The work by Wang Yanan et al. 2023. Ecological Indicators can be discussed to enrich this part contents.

**Response:** Thank you for the valuable suggestion. We believe that discussing the work by Wang Yanan et al. 2023, Ecological Indicators, can significantly enhance the content of this section. We have incorporated the following content: "For instance, PAR represents the portion of solar energy that can be utilized by plants and is an essential component in chloroplast reactions. PAR drives a nonlinear response of GPP to Solar-induced fluorescence (SIF) across different seasons, resulting in a strong positive correlation between GPP and SIF (Wang et al., 2023b)."

L391 Yuanyuan et al., 2018; L412 Y et al., 2022 these wrong citations need to be revised.

**Response:** Thank you for bringing these citation errors to our attention. We have revised the incorrect citations as follows: "Yuanyuan et al., 2018" has been corrected, and "Y et al., 2022" has been appropriately revised.

Once again, we express our sincere appreciation for your commitment to ensuring the quality and validity of scientific research. Your expertise and thoughtful suggestions have played a crucial role in refining our manuscript. We believe that the revised version now more effectively communicates the key findings and contributes meaningfully to the scientific discourse in our field.

---

## Author Comment (AC4)

Zhu and others study carbon flux in a subalpine forest on the Qinghai-Tibet Plateau. The interesting findings weren't discussed and the manuscript did little to inform readers about how carbon exchange in this system works. Rather, values were mostly compared against other studies, which was at times interesting, and many statements about global change that were not of particular reference to this study were made, which was distracting. The study, especially the Discussion needs to be comprehensively re-written to focus on the interesting findings of the study rather than a wandering review.

We thank the anonymous reviewer for providing her/his valuable feedback on our research. These professional insights are crucial for improving the manuscript, and we sincerely appreciate the thorough review. Replies to reviewers' comments are in blue below each comment. Line numbers refer to the line numbers after all revisions were made in the final manuscript without track changes on. The modified parts of the manuscript have all been marked in red font.

line 20 is both remarkably general ('autumn') and specific ('12:30'). reporting extremes is tricky because it may be an outlier; I would simply exclude this passage.

**Response:** Thank you for bringing this to my attention. We realize that describing the carbon flux in this way in flux studies was inappropriate, so we removed it.

23: it's already known that PAR is the most important variable if water or other factors aren't limiting. Can you say for certain that elevation is really the driver of carbon cycling or is it the climate characteristics that covary with elevation?

**Response:** Your point is precise, highlighting a deficiency in our analysis. We acknowledge that categorizing elevation as the influencing factor for local carbon exchange is inappropriate. Therefore, we have revised the analysis in this section, stating conservatively in the analysis of Figure 7 that the elevation of these sites. " According to existing results, an increase in elevation may lead to a reduction in carbon uptake …" . (in line 346)

line 29 and elsewhere: I have no idea where or when it was decided that eddy covariance measurements are accurate to 5 significant digits. '389' or perhaps better yet '390' is more realistic.

**Response:** Thanks for pointing this out. We have rounded the values to integers. (in line 27)

how was the statement on line 32 determined?

**Response:** In this study, we identified the carbon sink status of the subalpine forests on the Qinghai-Tibet Plateau. We recognized that the previous assertion was too categorical, as research on the carbon cycling of alpine forests in the vast Qinghai-Tibet Plateau is currently limited. Therefore, we have modified this statement to : "to positively influence global carbon cycling and promote "carbon neutrality and peak carbon," strengthening the protection and management of subalpine forests is crucial. Although our research has shown that these forest is currently playing a role in continuous carbon absorption, there are significant data gaps on the Qinghai-Tibetan Plateau. Therefore, it is essential to enhance continuous monitoring of forest carbon absorption processes in the future." (in line 33-38)

40: simply also note the year and location (assumably Mauna Loa)

**Response:** We appreciate your suggestion and have added the year and location to line 40. "in May 2021, a recorded peak of 419 parts per million (ppm) was observed at the Mauna Loa Observatory in Hawaii". (in line 43-44)

line 51-53: probably unnecessary to note in a study about forests in China.
**Response:** Thank you for your advice. In line 56, we have removed unnecessary citations and descriptions.

line 67 is inconsistent with a study that uses eddy covariance to measure carbon dioxide uptake.
**Response:** We have removed the reference. (in line 68)

line 90 is confusing because this paper doesn't quantify feedbacks (which is hard to do). Simply removing it is probably best.
**Response:** We have removed the reference and description.( in line 85)

133: there is no typical mountain climate.
**Response:** We have removed the redundant description. (in line 145)

would the density of air not be needed for equation 1?
**Response:** It is a theoretically simplified formula based on the eddy covariance system, which includes the molar concentration. relevant references have been cited here. (in line 177)

188: a storage flux of zero is not a safe assumption for a forest
**Response:** Your comments is appreciated. We have revised the measurement values and recalculated the results. (in line 203-210)

190: this isn't exactly the Michaelis-Menten model; rather it's a rectangular hyperbola, which has the same shape (if the vmax and Km parameters happen to be identical)
**Response:** Thanks for pointing this out. We have re-written the description in line 211 accordingly.

2.4: this section is usually called 'flux partitioning' or similar.
**Response:** We appreciate your suggestion, and we have modified section 2.4 to "Flux partitioning". (in line 237)

3.1 can be shortened considerably. There's too much unimportant text. Remove all unnecessary words and does a reader really need to know the maximum and minimum of things like RH?
**Response:** We have removed redundant descriptions here. (line 265-270)

245: this is qualitative; anything over the light saturation point is 'favorable for photosynthesis'
**Response:** We realized that the statement here was too conclusive. We have not added any analysis of data related to the intensity of photosynthesis. Therefore, we removed this description.

257: I struggle to see how Beijing time is relevant for a study in southwestern China. Using the solar zenith angle is probably more useful in this section. I don't see any evidence from the figures that

there's much of an afternoon drawdown; honestly this section mostly just says that carbon uptake follows light, which is already known. I recommend removing or dramatically simplifying.

**Response:** Thank you for your suggestions. The reason for using Beijing time in this context is that flux studies in these regions universally adopt Beijing time for consistency and ease of comparison across different studies. For the convenience of other readers' understanding, we have marked the time here as UTC+8. The previous description may have resulted in some redundancy, and we have simplified the content here with your suggestion. (in line 282-286)

Figure 4 requires an analysis of hysteresis, which would probably yield some interesting results. regression lines are not particularly useful here. Surprised to see that equation 3 wasn't used to study the response to PAR and that a linear model was used instead.

**Response:** Thank you for your suggestion. In the previous analysis, we employed a simple linear fit. We have taken the reviewers' advice and opted for PCA analysis to reveal more details. We have reanalyzed, plotted, and described figure 4. (in line 313)

Statements like 'The forest ecosystem respiration rate was lowest in winter and slightly higher in spring' are too obvious to really warrant mentioning, but differences in respiration between summer and autumn in different years are more interesting. The results section needs to be rewritten to focus on novelty.

**Response:** We removed the description you mentioned and modified the content here, adding an analysis of the differences in respiration between summer and autumn in different years. (in line 319-322)

297 and elsewhere: just remove everything to the right of the decimal places here.

**Response:** We rounded these numbers to integers.

section 3.6 was a surprise; previous text did not suggest that a synthesis would take place.

**Response:** We recognized that the content here might appear abrupt due to the lack of context in the preceding text. Therefore, we added the following description from lines 354-358: "To clarify the carbon sink potential of forests in the QTP and to compare it with other ecosystems, a search was conducted in two authoritative databases, Web of Science and China National Knowledge Internet, for research articles on the current utilization of EC systems in the QTP. A total of 82 research results were collected from 48 studies, and their annual average environmental factors, such as air temperature, precipitation, and altitude, were obtained."And changed the research objectives from lines 133-135 to: "3)    Since the carbon sink potential of forest ecosystems in the QTP is currently unknown, we evaluated the carbon exchange capacity of subalpine forests by comparing existing data with other ecosystems in the QTP."

Figure 7 isn't really carbon sequestration potential. It's observed C flux. fewer significant digits and units are necessary.

**Response:** Thank you for pointing this out. We agree that these values are real observed C fluxes rather than C sequestration potential. We have revised Figure 7, maintaining integer values for the significant digits in this section. (in line 354)

323-335 is too well known to warrant mentioning. What is it that is unique about the present study system? everyone knows that plants need light and proper temperatures. VPD is based on TA and RH. It is interesting to note how the study system was constrained by VPD.

**Response:** Your very pertinent suggestion is highly appreciated. We agree that this section contained unnecessary content. Considering other reviewers' suggestions to add more detailed information in this area, we have made comprehensive modifications. Following your advice, we have removed some content while retaining a portion of it. (in line 357-372)

337: this study isn't about fires and logging. Everything from 335 to 350 is expository material. It doesn't belong in the Discussion, and was a random assortment of references that was not organized very well.

**Response:** We removed the content in this section.

353: a discussion about why would be more interesting. Monson et al. (2006, GCB) and similar references covered this topic.

**Response:** Thank you for your suggestion. In the revised manuscript, we incorporated a discussion on the reasons behind the observed trends. We cited Monson et al. (2006, GCB) and similar literature to provide a more comprehensive understanding of the factors influencing the phenomena discussed in the paper. (in line 387-402)

remove lines 371-378.

**Response:** We removed the content in this section.

I absolutely cannot believe that the causes for differences in respiration in summer and autumn during different years was not discussed in the Discussion section, which is largely a poorly-organized narrative about different scientific studies.

**Response:** We fully appreciate your insights. In this revision, we added discussion about the differences in respiration during summer and autumn in different years (line 374-385), and we have enhanced the discussion by providing a detailed analysis of the possible reasons for this phenomenon.

We appreciate your guidance, and thank you once again for your efforts. We hope that our revisions meet your expectations and elevate the quality of the paper. Thank you once again for your valuable time and patience.

On behalf of all the authors,
Jinniu Wang,
E-mail: wangjn@cib.ac.cn

---

## Referee Report (RR1)

This is my second round of reviewing this manuscript. It seems to me that substantial modifications have been made in this resubmission and most of my suggestions have been incorporated. However, after reading through the manuscript, I find there are still some small issues that need to be addressed and some revisions require to be made. I list my concerns below and suggest a minor revision is needed to meet the publication requirement.

1. Line 18, please be specific about what 'pattern of higher rates' means.

2. Lines 21-22, PAR is an environmental factor rather than a climatic factor.

3. Line 22, should be more accurate to state that NEE is the net ecosystem $CO_2$ exchange.

4. Abbreviations should be used consistently and avoid repetition. For examples, in Line 78, the abbreviation of eddy covariance should be illustrated in Line 75; while in Line 76, the full name of NDVI should be given here rather than in Line 91; in Line 88, PAR has already been abbreviated in Line 84; the terms VPD and PAR appeared in Line 263 and Line 266 have already been abbreviated earlier. There should be other similar mistakes like the above mentioned, but I won't list all of them.

5. Lines 23-25, this result here is a bit incoherent, some introduction is needed to elucidate the sudden change from a site study to a spatial distribution.

6. Line 24, I understand that you observed higher NEE where there are higher air temperatures. However, according to the equation (y=-16.29x-86.67) in Figure 7, this correlation should be negative rather than positive. More importantly, the correlation is not that good given the correlation coefficient is only 0.17. I suggest rephrasing or reorganizing this sentence here to better show the rigorous results and innovative conclusions.

7. Line 26, the standard deviation or the range of NEE is needed to report here rather than a single mean value.

8. Lines 26-29, rephrase and refining are needed to make this sentence more concise. In addition, '368 g C m$^{-2}$'.

9. Line 29, 'This study provides …'. Besides, should avoid using the word 'essential' as this needs to be evaluated by the readers rather than the authors.

10. Line 30, should be 'alpine ecosystems' or 'sub-alpine ecosystems' rather than 'plateau ecosystems'.

11. In Lines 30-32, the proposition here is too broad and doesn't have that many direct connections with the research results of this study. I didn't see the necessity of stressing this here. Besides, the comma should be placed after the double quotation mark.

12. Line 36, 'The eddy covariance technique' or 'The eddy covariance method' is a better keyword here.

13. Line 62, the full name of the unit is not necessarily needed here.

14. Lines 99-101, it's better to merge the two sentences into one.

15. Lines 106-107, which are 'these ecosystems'?

16. Lines 118-120, it's better to rephrase this sentence to show that it's these kinds of research that is needed to be done rather than saying researchers should do this or that.

17. Line 124, what's the connection between this study and Yunnan-Kweichow Plateau. Maybe delete this part.

18. Lines 130-132, simply state your research aim directly, the reasons should be illustrated earlier in the introduction part.

19. Line 147, '… is around (should not be below) 30 meters….'?

20. Lines 149-152, better to merge the two sentences into one. Besides, should explain what 'southwest and southeast monsoons' means if you want to keep it.

21. Lines 152-156, refine and rephrase the two sentences.

22. Line 163, it should be that the EC system is deployed at the height of 35 m rather than the data were collected from this height.

23. Lines 169-170, what are the 'other environmental variables'?

24. Line 170, data was stored at 30-minute intervals.

25. Line 184, … $F_C$ raw data…

26. Lines 227-228, reformulate this sentence. It sounds like that only 27.33% of missing data were gap-filled. Should be that 27.33% of the data were filtered out and then the gaps were filled using Tovi based on Reichstein et al., 2005.

27. Lines 231-233, rephrase.

28. Lines 254, a supplementary file is needed to show the results that the authors compiled from the 82 sites and their locations and other environmental characteristics.

29. Line 293, should be specific that they are $CO_2$ fluxes.

30. Line 356, delete 'these factors'.

31. Line 384, … 9.03, 2.22, 2.71, ….

32. Line 403, … has indicated that ….

33. Figure 8, would be better to show the raw data points in this figure. It's hard to judge their relations based only on the four curves.

34. Lines 453-454, rephrase, the current description is confusing.

35. Lines 456-459, merge the two sentences and refine.

---

## Author Response (AR2)

Dear editors and reviewers,

We appreciate your professional comments and suggestions regarding our revised manuscript submitted to Biogeosciences. All the items have been discussed and addressed among co-authors. Additionally, we have addressed the minor issues you mentioned to ensure the manuscript meets the publication requirements. Please kindly see the responses as follows in details.

Response of suggestions for revision #1

This is my second round of reviewing this manuscript. It seems to me that substantial modifications have been made in this resubmission and most of my suggestions have been incorporated. However, after reading through the manuscript, I find there are still some small issues that need to be addressed and some revisions require to be made. I list my concerns below and suggest a minor revision is needed to meet the publication requirement.

Thank you very much for reviewing our manuscript again. We greatly appreciate your valuable feedback. We have made substantial revisions to the manuscript based on your comments. Here, we provide our responses to your comments as follows:

1. Line 18, please be specific about what 'pattern of higher rates' means.

**Response:** We have revised the sentence to "revealing the higher carbon dioxide exchange rates in summer and autumn and lower rates in winter and spring".(in line 19)

2. Lines 21-22, PAR is an environmental factor rather than a climatic factor.

**Response:** We have changed "climatic factors" to "environmental factors" in this section. (in line 22)

3. Line 22, should be more accurate to state that NEE is the net ecosystem $CO_2$ exchange.

**Response:** We have changed "net ecosystem exchange (NEE)" to "net ecosystem $CO_2$ exchange (NEE)" in this section. (in line 24)

4. Abbreviations should be used consistently and avoid repetition. For examples, in Line 78, the abbreviation of eddy covariance should be illustrated in Line 75; while in Line 76, the full name of NDVI should be given here rather than in Line 91; in Line 88, PAR has already been abbreviated in Line 84; the terms VPD and PAR appeared in Line

263 and Line 266 have already been abbreviated earlier. There should be other similar mistakes like the above mentioned, but I won't list all of them.

**Response:** Thank you very much for pointing out these errors. We have checked the abbreviations throughout the entire manuscript.

5. Lines 23-25, this result here is a bit incoherent, some introduction is needed to elucidate the sudden change from a site study to a spatial distribution.

**Response:** We have revised the first half of the sentence to: " In addition, we explored NEE and its influencing factors at the regional scale, found that air temperature …". (in line 26-28)

6. Line 24, I understand that you observed higher NEE where there are higher air temperatures. However, according to the equation (y=-16.29x-86.67) in Figure 7, this correlation should be negative rather than positive. More importantly, the correlation is not that good given the correlation coefficient is only 0.17. I suggest rephrasing or reorganizing this sentence here to better show the rigorous results and innovative conclusions.

**Response:** We have changed "found that the spatial distribution of NEE was significantly positively correlated with temperature" to " found that air temperature promotes carbon dioxide absorption (negative NEE values)". (in line 27)

7. Line 26, the standard deviation or the range of NEE is needed to report here rather than a single mean value.

**Response:** We have replaced this description with the annual average range of NEE values. (in line 29)

8. Lines 26-29, rephrase and refining are needed to make this sentence more concise. In addition, '368 g C m$^{-2}$'.

**Response:** Thank you for your suggestions. We have revised this sentence. (in line 30-32)

9. Line 29, 'This study provides …'. Besides, should avoid using the word 'essential' as this needs to be evaluated by the readers rather than the authors.

**Response:** We have revised the statement to: " This study provides valuable insights into the carbon cycling mechanism in sub-alpine ecosystems and the global carbon

balance". (in line 32-33)

10. Line 30, should be 'alpine ecosystems' or 'sub-alpine ecosystems' rather than 'plateau ecosystems'.

**Response:** We have changed "plateau ecosystems" to "subalpine ecosystems" in this section. (in line 33)

11. In Lines 30-32, the proposition here is too broad and doesn't have that many direct connections with the research results of this study. I didn't see the necessity of stressing this here. Besides, the comma should be placed after the double quotation mark.

**Response:** Following your suggestion, we have removed unnecessary descriptions from this section.

12. Line 36, 'The eddy covariance technique' or 'The eddy covariance method' is a better keyword here.

**Response:** We have changed "The eddy covariance system" to "The eddy covariance method" in this section. (in line 34)

13. Line 62, the full name of the unit is not necessarily needed here.

**Response:** We have removed the full name of the unit in this section. (in line 60)

14. Lines 99-101, it's better to merge the two sentences into one.

**Response:** We have combined the two sentences into: "Since the 1960s, the QTP has experienced a faster warming rate than lowland areas, a phenomenon projected to intensify by the end of the 21st century." (in line 97-98)

15. Lines 106-107, which are 'these ecosystems'?

**Response:** We have revised the sentence to: " discovered that by comparing carbon fluxes in ten high-mountain ecosystems with different grassland types, these ecosystems act as sinks for carbon dioxide". (in line 103-105)

16. Lines 118-120, it's better to rephrase this sentence to show that it's these kinds of research that is needed to be done rather than saying researchers should do this or that.

**Response:** We have revised the sentence to: " Long-term monitoring is necessary to understand how these forests will respond to climate change". (in line 116)

17. Line 124, what's the connection between this study and Yunnan-Kweichow Plateau. Maybe delete this part.

**Response:** Following your suggestion, we have removed "...and lies in the transitional zone between the QTP and the Yunnan-Kweichow Plateau". (in line 121)

18. Lines 130-132, simply state your research aim directly, the reasons should be illustrated earlier in the introduction part.

**Response:** We have revised the sentence to: " Evaluate the carbon exchange capacity of subalpine forests in the QTP by comparing existing data with other ecosystems in the region". (in line 126-127)

19. Line 147, '… is around (should not be below) 30 meters….'?

**Response:** We have changed it to "around". (in line 142)

20. Lines 149-152, better to merge the two sentences into one. Besides, should explain what 'southwest and southeast monsoons' means if you want to keep it.

**Response:** We have combined these two sentences. (in line 144-147)

21. Lines 152-156, refine and rephrase the two sentences.

**Response:** We have revised and rephrased these two sentences. (in line 147-151)

22. Line 163, it should be that the EC system is deployed at the height of 35 m rather than the data were collected from this height.

**Response:** We have revised the sentence to: " The EC system is deployed at a 35 m-high tower ".(in line 158)

23. Lines 169-170, what are the 'other environmental variables'?

**Response:** We have removed the redundant description here. (in line 164)

24. Line 170, data was stored at 30-minute intervals.

**Response:** We have changed "30 m" to "30-minute". (in line 165)

25. Line 184, … FC raw data…

**Response:** We have changed "FC raw valid data" to "FC raw data". (in line 178)

26. Lines 227-228, reformulate this sentence. It sounds like that only 27.33% of missing data were gap-filled. Should be that 27.33% of the data were filtered out and then the gaps were filled using Tovi based on Reichstein et al., 2005.

**Response:** We have revised the sentence to: "27.33% of missing data were interpolated using Tovi after filtering, resulting in a flux data set with complete data integrity ". (in line 221-222)

27. Lines 231-233, rephrase.

**Response:** We have modified the sentence to: "In terms of seasons, the average peak distances of the 90% flux contribution areas for winter, spring, summer, and autumn over the two years are as follows: 353.9, 358.2, 350.05, and 344.34m, respectively". (in line 225-227)

28. Lines 254, a supplementary file is needed to show the results that the authors compiled from the 82 sites and their locations and other environmental characteristics.

**Response:** "Thank you for your feedback, we have uploaded the supplementary file ".

29. Line 293, should be specific that they are $CO_2$ fluxes.

**Response:** We have changed "carbon fluxes" to "$CO_2$ fluxes" in this section. (in line 285)

30. Line 356, delete 'these factors'.

**Response:** We have removed "these factors" from this section. (in line 350)

31. Line 384, … 9.03, 2.22, 2.71, ….

**Response:** We have changed "9.025" to "9.03" in this section. (in line 378)

32. Line 403, … has indicated that ….

**Response:** We have changed "indicates that" to "has indicated that" in this section. (in line 397)

33. Figure 8, would be better to show the raw data points in this figure. It's hard to judge their relations based only on the four curves.

**Response:** Thank you for your feedback. We have displayed the raw data in the figure. (in line 406)

34. Lines 453-454, rephrase, the current description is confusing.

**Response:** We have revised the sentence to: " The research reveals that the subalpine forest acts as a carbon sink. Over the two years, the total NEE, GPP, and RE were −332, 1121, and 788 g C m$^{-2}$ in first year, and −351, 1199, and 847 g C m$^{-2}$ in second year. " (in line 447-449)

35. Lines 456-459, merge the two sentences and refine.

**Response:** We have revised the sentence to: "Combining results from other eddy covariance sites on the QTP, this study highlights that forests have the highest carbon sequestration potential, reaching 368 g C m$^{-2}$ annually, followed by meadows (-98 g C

$m^{-2}$), steppes (-64 g C $m^{-2}$), and shrubs (-61 g C $m^{-2}$). In contrast, wetlands were identified as a significant source of carbon dioxide (57 g C $m^{-2}$). " (in line 451-455)

Response of suggestions for revision #2

The manuscript as written represents an improvement, but the presentation is difficult to read as many paragraphs are quite long and touch on many topics, data quality in my opinion is overstated, the finding that PAR is the most important control over NEE on a half-hourly basis is obvious, and because temperature and RH contribute to the VPD calculation the PCA findings are interesting but could be teased out a bit more, it seems like because temperatures never get to high that the RH term of VPD is relatively more important in controlling plant function. This would be an interesting finding but is buried somewhat in an unnecessarily convoluted PCA. The study system is important and the manuscript makes a number of interesting findings, but comprehensive improvement would make it more valuable to the scientific community.

**Response:** Thank you for your detailed review and constructive feedback on our manuscript. We have taken your suggestions seriously and have made significant improvements to the manuscript. Specifically, we have revised the structure to improve readability, addressed the data quality concerns. We have revised some of the longer sentences that may have caused inconvenience to readers, making them more concise, and we have checked for grammar issues in the manuscript. We have presented the results more cautiously rather than exaggerating them. We have also added an analysis of the control of plant functions by relative humidity (RH) in the results section, and the contribution of RH to NEE is emphasized in the summary and conclusion. (in line25-26,295-297, 450)Your feedback has been very helpful to us, and we have carefully addressed the issues in the manuscript to improve it.

1.Equation 4 is written incorrectly (if using exp, don't superscript, just use e instead). Equation 8 is better but in many instances you needn't use the dot to represent multiplication.

**Response:** Thank you for your suggestion. We have corrected Formula 4 and replaced "dot" with "×" in the formula. (in line204, 216)

2.'The final flux data achieved a data integrity is 100%' isn't accurate. I'm not sure what 'integrity' means in this context and with a slope of 5% the nighttime data need to be interpreted very carefully.

**Response:** Here, we filled 27.33% of missing values and obtained a complete flux dataset. The wording here in the manuscript was difficult to understand, so we have revised it to '...resulting in a flux dataset with complete data integrity'. Errors caused by slope have been corrected in the manuscript through ' double-coordinate rotation".

3.Statements like that on line 455 can't be true; NEE would certainly exhibit significant differences across seasons.

**Response:** Thank you for pointing out the issue. Our previous statement was redundant. We have changed it to "NEE reached its peak in autumn." (in line451)

We appreciate your guidance, all the modified contents are marked in red in the manuscript, thank you again for your thorough review and constructive suggestions.

On behalf of all the authors,

Jinniu Wang,

E-mail: wangjn@cib.ac.cn